# Comparative Evaluation of Diagnostic Methods for Subclinical Benign Prostatic Hyperplasia in Intact Breeding Male Dogs

**DOI:** 10.3390/ani14081204

**Published:** 2024-04-17

**Authors:** Tomas Laurusevičius, Jūratė Šiugždaitė, Nomeda Juodžiukynienė, Sigita Kerzienė, Lina Anskienė, Vaiva Jackutė, Darius Trumbeckas, Ann Van Soom, Florin Petrisor Posastiuc, Henrikas Žilinskas

**Affiliations:** 1Kaunas Veterinary Practice, Veiveriu Str. 176a-2, 46415 Kaunas, Lithuania; 2Faculty of Veterinary Medicine, Veterinary Academy, Lithuanian University of Health Sciences, Tilžes Str. 18, 47181 Kaunas, Lithuania; jurate.siugzdaite@lsmu.lt (J.Š.); nomeda.juodziukyniene@lsmu.lt (N.J.); sigita.kerziene@lsmu.lt (S.K.); lina.anskiene@lsmu.lt (L.A.); v.jackute@gmail.com (V.J.); henrikas.zilinskas@lsmu.lt (H.Ž.); 3Department of Urology, Urology Clinic, Lithuanian University of Health Sciences, Eiveniu Str. 2, 50161 Kaunas, Lithuania; darius.trumbeckas@lsmu.lt; 4Department of Internal Medicine, Reproduction and Population Health, Faculty of Veterinary Medicine, Ghent University, Salisburylaan 133, 9820 Merelbeke, Belgium; ann.vansoom@ugent.be (A.V.S.); florin.posastiuc@ugent.be (F.P.P.); 5Department of Clinical Sciences, Faculty of Veterinary Medicine, University of Agronomic Sciences and Veterinary Medicine of Bucharest, 105 Blvd. Splaiul Independentei, 050097 Bucharest, Romania

**Keywords:** canine andrology, BPH, ultrasonography, rectal palpation, CPSE

## Abstract

**Simple Summary:**

Benign prostatic hyperplasia is a common condition in older intact male dogs, and it is often initially lacking in noticeable clinical signs. Due to this asymptomatic nature, it is frequently overlooked in clinical practice. Our study aimed to develop a simplified non-invasive examination plan for the early detection of subclinical benign prostatic hyperplasia in older intact dogs. We established a clinical rectal-scoring system and new diagnostic thresholds using non-invasive methods such as ultrasonography and analysis of canine prostatic-specific esterase from blood serum samples. Our study revealed elevated values of canine prostatic-specific esterase, increased prostatic volume, and higher values of prostatic artery vascular velocities in dogs with subclinical BPH. Additionally, we found a positive relationship between prostatic volume and the age and weight of the dogs. In summary, our findings indicate that multiple non-invasive examinations can aid in identifying or suspecting early stages of BPH based not only on proposed new thresholds for prostatic volume and canine prostatic-specific esterase, but also elevated prostatic artery vascular velocities, changes in size, shape, and echostructure of the prostate gland, and differences in rectal examination. This information enables practitioners’ timely intervention, preventing the progression of prostatic manifestations.

**Abstract:**

Benign prostatic hyperplasia (BPH) is an androgen-related non-neoplastic enlargement of the prostate gland that commonly affects both reproductive capabilities and the general health of intact dogs. The subclinical form of BPH can be challenging to diagnose due to a lack of clinical signs, even if rectal palpation is performed. Left untreated, this condition poses risks to the dogs’ health and breeding status. This study, involving 65 male dogs, aimed to investigate subclinical BPH through rectal palpation, ultrasonography, and analysis of canine prostatic-specific esterase (CPSE). Of the participants, 35 had subclinical BPH, and 30 served as a healthy control group. Dogs suspected of subclinical BPH, as determined by examination results from ultrasonography and CPSE analysis, underwent fine needle aspiration (FNA) guided by ultrasound to enhance diagnostic precision. Findings revealed distinct differences in rectal palpation and ultrasonography between subclinical BPH and healthy dogs. This study established diagnostic thresholds based on prostatic volume and CPSE values and proposed new thresholds for subclinical BPH. Additionally, results showed that prostate gland volume depended on the weight and the age of the dog. In conclusion, early detection of this condition is possible through various examinations, such as changes in ultrasound features, CPSE levels, and rectal palpation. All together, these methods can aid practitioners in early detection of BPH and assist with scheduling screening programs for dogs, ultimately promoting their overall health and reproductive well-being. In conclusion, we advocate for routine, non-invasive prostate screenings in breeding males, underlining the effectiveness of a combination of various multiple techniques for early subclinical BPH detection.

## 1. Introduction

The prostate gland is the only sex gland in the male dog’s reproductive tract [1]. Prostatic disorders significantly affect both the reproductive system and the overall health of canine patients. Among these, benign prostatic hyperplasia (BPH) is the leading condition diagnosed in intact dogs [2,3]. According to Socha et al. (2018), dogs of large and giant breeds often exhibit a propensity for developing BPH [4]. Linked to the aging process, this spontaneous lesion involves both an increase in cell numbers (hyperplasia) and an enlargement of individual cells (hypertrophy), typically manifesting in older dogs [5]. Despite that, it is important to note that BPH can be evident in dogs as young as 2 years old [6]. Moreover, it should be emphasized that in its early stages, this condition can remain asymptomatic until an increased glandular volume leads to the emergence of BPH-related symptoms such as constipation, hematuria, dysuria, changes in fertility status, and issues like excessive licking of the penis or lameness of the hind legs; in BPH-affected dogs, the prostate gland is more susceptible to bacterial infections, and if left untreated, such conditions such as prostatitis and prostatic cysts with the possibility abscess formation can develop [7].

Furthermore, the presence of BPH, prostatitis, or prostatic neoplasia have been linked to the onset of perineal hernias in intact dogs [8]. Regarding the reproductive health of dogs with BPH, there can be a decreased libido, issues with the ejaculatory process, and a noted decline in semen quality attributed to changes in prostatic fluid composition, which can lead to an elevated level of sperm cell DNA fragmentation [9]. Due to the fact that BPH presents no clinical signs in many cases, its diagnosis can be challenging. In regard to this, multiple examinations are recommended to ensure a reliable diagnosis [10,11]. A comprehensive diagnostic approach includes a full patient history, rectal palpation, evaluation of the organ using diagnostic imaging tools, semen collection, and culture and cytology of prostate tissue or fluid samples. Moreover, the levels of a distinct biomarker for prostatic disease, canine prostate-specific esterase (CPSE), is now highly recommended [12]. 

In clinical practice, rectal palpation is one of the methods to estimate the size, shape, flexibility, anatomical location, glandular surface, and sensitivity of the prostate gland [13]. While rectal palpation primarily provides access to the dorsocaudal regions of the prostate gland, it can pose challenges in small dogs due to their size or in giant breeds where the prostate may be positioned too cranially. Despite these limitations, it is still recommended to include this examination in routine procedures for dogs whenever possible, as it can serve as a valuable guide for further investigations if necessary [1,5,14,15,16].

In terms of using different diagnostic imaging methods, X-ray imaging of the prostate gland can be used to determine the location and size of the organ. However, it has a limited value due to interference of other abdominal organs, resulting in poor contrast images affecting the visualisation of the organ [17]. 

In regards to the diagnostic limitations of the X-ray, ultrasonography is proposed as a valuable method for examining the prostate gland [18]. 

Transrectal ultrasound, while the preferred approach for achieving precise images of the target organ, presents challenges in practicality due to the necessity of specialized equipment and sedation for the patient [19].

Abdominal ultrasonography is the preferred method over transrectal examination in most veterinary practices. Recognized as the most effective technique for prostate gland assessment, it enables the evaluation of the gland’s size, shape, tissue structure, and vascular velocities of the organ [10,20,21]. Furthermore, with the assistance of ultrasonography, biopsies and cytology samples can be acquired, as tissue analysis is the most accurate method for diagnosing prostatic disorders [22].

Interestingly, recent studies have highlighted the potential of CPSE as an innovative, non-invasive diagnostic marker for the assessment of prostatic status in dogs [20]. CPSE, classified as an arginine esterase, is synthesized by the columnar epithelial cells within the canine prostate gland, with its secretion increasing as benign prostatic hyperplasia progresses, accompanied by hypertrophic and hyperplastic changes in the prostatic epithelium. Due to this fact it has been shown that dogs diagnosed with BPH have elevated CPSE levels [23,24]. However, a concrete diagnosis of prostatic conditions can be achieved with ultrasound-assisted fine needle aspiration (FNA) or a biopsy of the prostatic tissue [25]. 

The aim of this study is to develop a simplified approach for the early detection of subclinical BPH in older intact dogs, avoiding invasive diagnostic procedures. This involves establishing new diagnostic thresholds and clinical scoring systems based on non-invasive techniques such as digital rectal examination, ultrasonography, and analysis of CPSE. 

## 2. Materials and Methods

In this investigation, a cohort of 65 client-owned intact breeding male canines, representing large and giant breeds, was assembled. The subjects encompassed a spectrum of ages and weights, with details provided in Appendix A (Table A1 and Table A2) regarding breed classification and age distribution. Dogs were randomly selected from those undergoing routine health check-ups at the veterinary practice. During these examinations, non-invasive procedures such as general clinical examination, rectal palpation, and ultrasonography of the abdominal and chest cavities were performed. However, dogs displaying ultrasonographic alterations of the prostate gland, such as intraprostatic cysts, asymmetrical shape, heterogeneous echotexture, and increased prostate gland vascular velocities, were recommended for further investigation to diagnose subclinical BPH. Blood samples were collected from dogs exhibiting BPH-related alterations of the prostate gland for CPSE concentration analysis. Dogs with a CPSE concentration reaching or exceeding ≥61 ng/mL, alongside the aforementioned sonographic alterations, underwent ultrasound-assisted FNA to accurately confirm the diagnosis of subclinical BPH. The CPSE concentration threshold of ≥61 ng/mL and the ultrasonographic alterations related to BPH were determined based on the studies conducted by Pinheiro et al. (2017) and Cunto et al. (2022), respectively [14,26]. Ultrasound-guided FNA was performed based on techniques published by Kustritz et al. (2006) [27]. After cytological analysis, two study groups were conducted: the healthy dogs group (n = 30) and the subclinical BPH-affected dogs group (n = 35). The healthy group had an average age of 3.7 years (range: 3 to 6 years) and an average weight of 39.8 kg (range: 27 to 58 kg), while the subclinical BPH-affected group had an average age of 6.9 years (range: 4 to 10 years) and an average weight of 40.1 kg (range: 21 to 60 kg).

### 2.1. Clinical History and Examination

None of the dogs had any documented history of compromised semen quality or any other reproductive tract disease, nor did they have any general health disorders within the 6 months prior to the study. Additionally, no unsuccessful matings or pregnancies were recorded. None of the dogs included in this study were mated in the period of 6 months since the beginning of the study. Initially, a comprehensive breeding soundness evaluation was performed by a practitioner specializing in the field of small animal reproduction. This evaluation included visual and morphological inspections of the prepuce, penis, and testes, along with an additional sonographic examination of both testicles. Based on the results of all aforementioned examinations, all dogs were determined to be clinically healthy without any signs of prostatic or testicular diseases. 

### 2.2. Rectal Palpation

Rectal palpation was conducted consistently by the same individual across all dogs in a blinded manner. Dogs were posed in a standing position and gently restrained. Using a lubricated gloved finger, the examiner palpated the anterior rectal wall to assess the prostate gland. In instances where the prostate gland was not palpable, the operator applied gentle pressure with his left hand to the abdominal wall, maneuvering the gland into a dorsocaudal position until it became palpable. Assessment criteria included the prostate gland’s size, shape, consistency, surface, and position. Additionally, a pain and size score system was suggested and employed for a comprehensive evaluation. The pain score system comprised 4 scoring points:Score 0: no reaction to palpation. The dog remains calm without muscle tensing, vocalization, or attempts to move.Score 1: mild discomfort. Slight tensing of abdominal muscles or flinching during palpation, but no vocalization or significant movement.Score 2: moderate discomfort. Clear flinching, pulling away, or trying to sit down during the examination. May show signs of discomfort like turning the head to the examiner.Score 3: severe discomfort or pain. Vocalizing during palpation, strong attempts to pull away or aggressive behaviour due to pain.

Based on size evaluation, a 4-point scoring system was employed to assess the degree of prostate gland occupation within the rectal lumen.

Score 0: prostate gland is barely palpable within the rectum.Score 1: prostate gland is palpable, occupying a small portion of the rectal lumen.Score 2 prostate gland is palpable, occupying a moderate portion of the rectal lumen.Score 3: prostate gland is palpable, occupying a significant portion of the rectal lumen.

### 2.3. B-Mode Ultrasonography

The ultrasound examination was performed by a specialist in small animal diagnostic imaging. A Mindray Vetus-7 (Mindray Bio-Medical Electronics Co., Ltd. Shenzhen, China) ultrasound machine was used. The dogs were placed in dorsal recumbency using soft positioning aid beds. The prostate gland was scanned with a microconvex probe in transverse and longitudinal planes. The frequency range was set between 5.0 and 7.5 MHz. Sonographic images of glandular parenchyma were categorized as homogenic or heterogenic based on its appearance. Intraprostatic cysts were determined as anechoic, roundish areas of different sizes. No variations in cyst size or volume were distinguished. The region of interest (ROI) of the prostate gland was drawn using a freehand tool navigator to highlight the borders of the prostate gland in the transversal plane. Prostatic dimensions were measured as follows: prostatic length (L) on the longitudinal plane and prostatic width (W) and height (H) on the transversal plane. The measurements of height and length included both lobes of the prostate, and the average value in centimeters was calculated, whereas prostatic length was defined as the maximum diameter along the urethral axis and measured in centimeters (Figure 1). The prostatic volume (PV) was calculated using the ellipsoid body formula proposed by Ruel et al. (1998): PV (cm) = (L × W × H) × 0.523 [18].

The color Doppler technique was employed to examine the blood flow in the prostatic artery (*a.prostatica*) marginal and subcapsular locations, following the method suggested by Zelli et al. (2013) [21]. In the pulse wave (PW) mode, the sample volume was set to cover the entire lumen of a vessel, ensuring that waveforms representing at least four consecutive cardiac cycles were recorded. Specific blood flow parameters, including peak systolic velocity (PSV), end diastolic velocity (EDV), and resistance index (RI) were calculated. The ultrasound inbuilt algorithm package automatically derived these values for each waveform. The values from three sweeps were then averaged to determine a mean value for each parameter at each location. Additionally, color gain adjustments were made to mitigate flash artifacts.

Subsequently, ultrasound-guided FNA was conducted. The cytology samples were assessed by a clinical pathologist (Figure 2). BPH was identified through the characteristics of the cytological samples: the presence of extensive groups of epithelial cells with either columnar or polygonal morphology, a minimal nucleus-to-cytoplasm ratio, and uniform round nuclei showcasing small nucleoli alongside finely granulated chromatin patterns [25,28].

### 2.4. Canine Prostatic-Specific Esterase 

Blood samples were collected from all study dogs by inserting a 21-gauge needle into the cephalic vein. VACUETTE™ Z serum blood collection tubes (Thermo Fisher Scientific Inc., Waltham, MA, USA) were chosen. After collection, samples were promptly sent to the laboratory for analysis. The concentration levels of CPSE were analyzed within 30 min according to the guidelines of the Speed™ Reader (Virbac, Carros, France), a laser-induced fluorescence immunochromatographic serum analyzer.

### 2.5. Ethical Statement 

The research was performed in accordance with the Law of the Republic of Lithuania No. VIII-500 on the Care, Welfare and Use of Animals, dated 6 November 1997 (“Valstybės žinios” (Official Gazette) No. 108, 28 November 1997) and orders of the State Veterinary Service of the Republic of Lithuania on the Breeding, Care and Transportation of Laboratory Animals (No. 4-361, 31 December 1998) and on the Use of Laboratory Animals for Scientific Tests (No. 4-16, 18 January 1999). The approval number for this study was PK No. 012856. Informed consent was signed by every owner and obtained as additional approval for clinical examinations and procedures.

### 2.6. Statistical Analysis

Statistical analyses were conducted using IBM SPSS Statistics 29.0.0.0 (241). The normality of the datasets for prostate size and CPSE values was assessed employing the Shapiro–Wilk test. Calculations of means and standard deviations were performed for the prostate size and CPSE values within the subclinical BPH-affected and healthy groups. The statistical significance of the observed differences was determined using a Student’s *t*-test for independent samples.

Receiver operating analysis (ROC) was executed to ascertain the cut-off values for prostate gland volume and CPSE values, establishing precise thresholds indicative of subclinical BPH occurrence. Disparities in prostatic rectal palpation results between the subclinical BPH-affected and healthy groups were statistically evaluated utilizing the χ2 test, with a Bonferroni correction applied. A significance level of *p* < 0.05 was considered to denote statistical significance in all analyses.

## 3. Results

### 3.1. Rectal Palpation

In the healthy group, rectal palpation primarily revealed typical prostate gland characteristics. The gland was prominently located in the cranial section of the pelvic inlet, displaying a symmetrical appearance with a uniform, smooth, and elastic texture. The size of the organ did not occupy a significant portion of the rectal lumen, and the evaluation scores were mostly 0–1.

Upon examining dogs affected by subclinical BPH, notable changes were observed through rectal palpation, particularly in the shape, size, and consistency of the prostate. There was a discernible shift in the positioning of the prostate gland, which leaned more cranially towards the pelvic brim. Furthermore, in this group, approximately half of the prostate surfaces exhibited a rough texture. Based on size, more than half of the prostate glands in this group exhibited enlargement, resulting in moderate occupation of the rectal lumen. Detailed results of prostate gland rectal palpation are presented in Table 1.

### 3.2. Ultrasonographic Evaluations of Prostate Glands

Within the subclinical BPH-affected dogs, 20% (n = 7) displayed glandular asymmetry, while in the healthy group, 16.6% (n = 5) displayed a similar shape. In terms of tissue structure, our observations revealed a prevalent heterogeneous pattern in the BPH study group, with 85.7% (30 out of 35) of dogs exhibiting this feature. In contrast, only 13.3% (4 out of 30) of dogs in the healthy group demonstrated heterogeneous prostatic tissue. However, the statistical analysis revealed that the relationship between echogenicity and the shape of the prostate was not found to be statistically significant in either the affected group, the healthy group, or when considering both groups collectively (*p* > 0.05).

In terms of intraprostatic cyst occurrence, our analysis showed a higher prevalence within the subclinical BPH group, where 85.7% (n = 30) of cases exhibited a diffuse cystic pattern. In contrast, among the healthy dogs, only 10% of cases (n = 3) exhibited intraprostatic cysts within the prostatic parenchyma. We further conducted a statistical analysis to assess the correlation between the presence of intraprostatic cysts and the shape of the prostate gland. Our findings indicated that the observed correlation was not statistically significant in either the subclinical BPH group or the healthy group, nor when assessing both groups combined.

This study identified differences in prostatic dimensions between subclinical BPH-affected dogs and the healthy group. Specific measurements for the subclinical BPH group included a prostatic length with mean ± SD values of 5.34 ± 1.29 cm, width of 5.05 ± 1.17 cm, and height of 4.12 ± 0.94 cm. In contrast, the healthy group measurements stood at 3.57 ± 0.77 cm for length, 3.70 ± 0.94 cm for width, and 3.45 ± 0.88 cm for height. Statistical analysis showcased significant differences in prostatic length, width, and height of the prostate gland between both study groups (*p* < 0.01).

Furthermore, the subclinical BPH group demonstrated larger prostatic volume, with mean ± SD values of 64.51 ± 43.62 cm^3^, in contrast to the healthy group, which had a prostatic volume of 26.93 ± 17.93 cm^3^ (*p* < 0.001).

Additionally, we examined the relationship between prostatic volume and age across both study groups. Analyzing the healthy control and subclinical BPH-affected dogs collectively, a moderate correlation between age and prostate volume was evident (*p* < 0.001). Yet, within the subclinical BPH-affected dogs group, the correlation between age and prostate volume was statistically insignificant (*p* > 0.05). In this group, the prostate volume will rise by an average of 2.29 cm^3^ for each year increase in age (*p* = 0.597). The regression analysis for the healthy control group indicates that with each year of age, the prostate volume will increase on average by 2.74 cm^3^ (*p* = 0.527). While these changes within each group are statistically insignificant, when considering both the healthy and subclinical BPH-affected groups together, a notable moderate correlation between age and prostate volume is observed (r = 0.423, *p* < 0.001). The results are shown in Figure 3.

Regarding the correlation between weight and prostate volume in the healthy group, a moderate correlation was observed (r = 0.475, *p* < 0.01). For each kilogram increase in weight, there was an average prostate volume increment of 0.82 cm^3^ (*p* < 0.01). Conversely, in the subclinical BPH-affected group, this relationship was faint (r = 0.217, *p* = 0.209), a trend similarly noticed when evaluating the whole study cohort (r = 0.240, *p* = 0.055). The results are detailed in Figure 4.

A receiver operating characteristic (ROC) analysis was employed, emphasizing the value of prostate volume in distinguishing healthy dogs from those afflicted with subclinical BPH. The area under the curve (AUC) was computed to be 0.87 with a notable cut-off value set at 35.16 cm^3^ (*p* < 0.001). This analysis underscores the diagnostic potential of assessing prostate volume. An ROC analysis graphical image is shown in Figure 5.

### 3.3. Color Doppler Evaluation 

A comparative analysis between the two study groups unveiled notable distinctions in color Doppler assessments at different locations along the prostatic artery.

Both the PSV and EDV were notably higher in the BPH group across both locations. Furthermore, the RI was elevated in the BPH group in both the marginal and subcapsular locations. However, statistical differences in RI were observed only in the marginal location (*p* < 0.01). Detailed data and statistical representations of the Doppler parameters for each group can be found in Table 2.

### 3.4. CPSE Analysis

Our investigation highlighted substantial differences in CPSE values between the subclinical BPH-affected and the healthy dog groups. Specifically, the mean ± SD values were recorded as 38.85 ± 14.55 ng/mL (range from 17.53 to 67.8 ng/mL) for the healthy group and escalated to 203.3 ± 90.39 ng/mL (range from 97.31 to 487.54 ng/mL) for the BPH group (*p* < 0.001). These data are graphically elucidated in Figure 6.

Our data underscored the feasibility of using CPSE values as a determinant to differentiate healthy dogs from those with BPH. The AUC was a perfect 1.00 (*p* < 0.001), with a cut-off threshold set at 82.56 ng/mL. Thus, when the CPSE metric exceeds or equals 82.56 ng/mL, it becomes feasible to pinpoint dogs showing an asymptomatic form of BPH with a confidence level of 100% (*p* < 0.001). The results are shown in Figure 7.

## 4. Discussion

### 4.1. Rectal Palpation

Clinically, the health of the prostate gland is often gauged using digital rectal palpation, which evaluates various factors, including size, shape, position, consistency, and pain, among others [11,13]. The results obtained from our study corroborate and extend the findings reported by other authors, providing valuable insights into the rectal palpation of canine prostates in both healthy and subclinical BPH-affected groups. 

The normal prostate is smooth and symmetrical in shape, and it is free of pain on digital examination. The dorsal sulcus of the prostate is easily palpable and can be a useful landmark for those with limited experience [5]. In our study, the healthy group predominantly exhibited standard prostate characteristics, characterized by a cranially situated gland with symmetrical lobes, a smooth and elastic texture, and an evident urethral groove, with size scores predominantly falling within the range of 0–1. The majority of subjects demonstrated minimal discomfort during rectal palpation.

In dogs with BPH, the prostate can be shifted cranially into the abdomen. Each of the two lobes of the prostate should be symmetric in size and shape. The consistency should be firm, but not hard. Gentle palpation of the prostate should not be painful [22]. Our results align with these observations, indicating a noticeable cranial shift in the positioning of the prostate gland upon examination of dogs with subclinical BPH. In some cases, mild discomfort was noted, but no severe pain reactions were observed. Additionally, approximately half of the cases exhibited a rough texture on the prostate surface, possibly indicative of an expressed cystic pattern in the gland, which can lead to surface texture irregularities. 

According to the size scoring system, more than half of the dogs in the group were classified with a score of 3, indicating mild prostatomegaly. Such enlargement is frequently observed in the subclinical stage of BPH, often presenting without overt clinical signs such as tenesmus, diarrhea, or stranguria [11]. 

However, an asymmetrical shape was observed in over 80% of dogs in the group, potentially indicating the progression of the current condition or the presence of an intraprostatic cystic pattern. It is well-established that asymmetry of the prostate gland can signify neoplastic or inflammatory processes [16]. Nevertheless, cytological examinations did not reveal evidence of either condition. This result may be attributed to operator variability, as rectal palpation is highly dependent on the skill and technique of the examiner.

Nonetheless, in general, rectal palpation remains a valuable tool, especially in cases like acute prostatitis, where immediate management is crucial, and prostatic neoplasia, where notable changes in shape, size, and pain are present, making it highly recommended for use in intact dog patients [13].

### 4.2. Ultrasonographic Evaluation

Building upon the findings of Mantziaras et al. (2017), which underscored the significance of early detection of prostatic abnormalities in dogs, our investigation delved deeper into this recommendation. In line with their study, which involved a substantial cohort of 1003 dogs from various breeds categorized by life expectancy, we aimed to elucidate the optimal age for preventive ultrasonographic examination of the prostate [29]. Confirming their assertion that around 40% of expected longevity is a critical period for detection of prostate gland alterations, our study included a healthy group with an average age of 3.7 years and a subclinical BPH-affected group with an average age of 6.9 years. This age discrepancy between groups serves to highlight the importance of age-related factors in prostatic health assessments. While the control group’s younger age suggests minimal susceptibility to prostatic changes, the inclusion of the BPH-affected group, despite their lack of symptoms, aligns with the recommended lifespan limit identified by Mantziaras et al. (2017).

Based on our results, the prostate glands in the subclinical benign prostatic hyperplasia (BPH) group primarily exhibited symmetrical shapes and heterogeneous tissue structures, characterized by a diffuse cystic pattern. In contrast, healthy dogs displayed different findings, with predominantly homogeneous tissue patterns and an absence of intraprostatic cysts. However, the shape of the prostate gland was symmetrical in most cases.

A recent study by Nizanski et al. (2020) on 10 dogs diagnosed with BPH, with a mean age of 9.5 (SD ± 3.5) years and a mean weight of 14.12 (SD ± 12.17) kg, reported similar ultrasonographic findings. In this group, the prostate gland appeared as heterogeneous with focal lesions less than 1 cm in diameter (intraprostatic cysts). Our results correspond to the findings mentioned above, reinforcing the consistency of ultrasound characteristics in dogs with BPH across different studies. Similar findings were described in the study by Russo et al. (2012), where eight dogs (age 5.9 ± 3.2 SD years and weight 19–37 kg) diagnosed with BPH exhibited increased tissue echogenicity and the presence of intraprostatic cysts in all cases [30].

Regarding prostatic dimensions, our study indicated that in subclinical BPH-affected dogs, the length, width, and height of the prostate gland were higher compared to their healthy counterparts. In the study by Ruel et al. (1998), which investigated 100 intact healthy dogs with a mean age of 5.1 (SD ± 3.4) years and a mean weight of 18 (SD ± 11.8) kilograms, the mean SD± values for prostatic length, width, and height were 3.4 ± 1.1 cm, 3.3 ± 0.9 cm, and 2.6 ± 0.7 cm, respectively [18]. These results align with our findings for healthy male dogs. However, it is important to note that in the subclinical BPH group, these values were significantly higher, indicating prostate gland enlargement in this condition.

In a study by Pasikowska et al. (2015), where computed tomography was used to assess the dimensions of the prostate gland, the results from 20 intact dogs diagnosed with BPH, ranging between 15 and 45 kg and 5–11 years old, showed a prostatic length, width, and height of 4.38 ± 1.1 cm, 4.89 ± 0.87 cm, and 4.49 ± 0.94 cm, respectively [1]. Similar values were found in our study using ultrasonography. These findings also align with the fact that prostate gland volume is higher in BPH-affected dogs compared to healthy ones, as published in the recent study by Genov et al. (2021) [31].

In the realm of veterinary research, several studies have consistently described a direct correlation between prostatic volume, age, and weight [18,19,32]. Such a trend, broadly accepted in the field, suggests that as dogs age, there is an expected increase in prostatic volume, which might be further influenced by their weight. However, our current investigation paints a slightly different picture, prompting a reevaluation of these established notions.

Upon investigating a sample of 65 male dogs, our findings revealed a nuanced relationship. While there was an undeniable significant correlation between prostatic volume, age, and weight on a holistic scale, subgroup analyses unveiled certain inconsistencies. The relationship between volume, age, and weight seemed to fluctuate depending on the specific groups considered.

A potential explanation for these divergent results might stem from the clinical progression of prostatic conditions in our sample. No dog in our study showcased overt clinical signs related to prostatic enlargement. If we operate under the premise that only substantial enlargement of the prostate gland triggers clinical manifestations, then the observed positive correlation in specific subgroups starts making intuitive sense.

Such unexpected findings offer a fresh perspective and emphasize the significance of comprehending the nuances of prostatic health not only in the subclinical stage but also in the broader context of trends in intact dogs.

In a study focusing on beagles, Wheaton et al. (1979) reported a range of prostatic volume deriving from 20 to 31 cm^3^ in BPH-affected males [33]. In a 2000 study by Kamolpatana et al., 12 intact male dogs, all under 5 years old and with an average weight of 21.8 kg, were examined using ultrasound. The mean ±SD prostatic volume values found in this study were 16.77 ± 11.77 cm^3^ [34]. In comparison to our study, we found a slightly higher average prostatic volume of 26.93 ± 17.93 cm^3^ in the healthy dogs group. Compared to the study by Kamolpatana et al. (2000), our healthy dogs exhibited some distinctions, being, on average, 3.7 years old with an average weight of 39.8 kg.

In the investigation conducted by Ruel et al. (1998), 100 intact healthy male dogs of various breeds, ages, and weights were examined, and prostatic volume was calculated utilizing ultrasonographic measurements and the ellipsoid body formula. The mean standard deviation of prostatic volume across all dogs was found to be 18.9 ± 15.5 cm^3^ [18]. These divergences might shed light on the disparities observed in prostatic volume ranges between our study and the previously discussed research by both Kamolpatana et al. (2000) and Ruel et al. (1998). Nonetheless, neither of the two aforementioned studies examined the precise health status related to prostatic volume.

In the research conducted by Hosseinpour et al. (2022), 24 intact male dogs with an average age of 7.6 years and an average weight of 13.6 kg, all exhibiting clinical signs associated with BPH, were examined using an ultrasound, and their prostatic volume was calculated. The mean and standard deviation values were 14.32 ± 12.62 cm^3^ [2]. In contrast, our study demonstrated that, in dogs (average age 6.9 years and average weight 40.1 kg) affected by subclinical BPH, the mean ± SD values of prostatic volume were notably higher, registering at 64.51 ± 43.62 cm^3^. The variance in results might be attributed to differences in sample size, weight, and age between the studies; specifically, our research focused on large breed dogs, while Hosseinpour et al. (2022) exclusively included small breed dogs. Our results underscored a markedly elevated prostatic volume in dogs affected by subclinical BPH, with the diagnosis being validated through cytological examination. This increased prostatic volume is attributed to the prostate enlargement that is intrinsically associated with the condition.

In the study conducted by Dearakhshandeh et al. (2020), twenty-five male intact mixed-breed dogs, aged between 1 and 3 years and weighing 15–20 kg, were employed to investigate the induction of BPH, which involved injections of testosterone enanthate and estradiol benzoate. Ultrasound examinations revealed that the prostate volume of the induction group exhibited a notable increase in prostate volume from 9.66 ± 4.81 cm^3^ on day 0 to 20.59 ± 6.83 cm^3^ on day 63 [35]. While there are discrepancies in the age and weight of the dogs used in our study, the results broadly mirror our own, highlighting that in instances of subclinical stage of BPH, prostate gland volume is significantly elevated compared to healthy dogs.

Additionally, in our research, the ROC analysis suggested a prostatic volume threshold for subclinical BPH in dogs to be above 38 cm^3^. However, it is crucial to note that our study predominantly involved large to giant breed dogs, with an average age of 6.9 years and an average weight of 40.1 kg. This can potentially explain the variations observed in previous studies, where some studies excluded variations based on breed or assessed only the prostate gland volumes in healthy intact dogs and not in neutered dogs [18,26,32,34,36].

### 4.3. Color Doppler Evaluation

Our research emphasizes the considerable potential of color Doppler in this domain. We compared our study results with the one that was conducted by Zelli et al. (2013) [21]. In their work, the average values for PSV, EDV, and RI in the marginal prostatic artery were 33.23 ± 2.29 cm/s, 6.14 ± 0.71 cm/s, and 0.85 ± 0.03, respectively. In contrast, our study showed values of 22.29 ± 1.4 cm/s, 4.44 ± 0.33 cm/s, and 0.80 ± 0.02 for the same metrics. 

Exploring the subcapsular margins, Zelli et al. (2013) reported PSV, EDV, and RI values as 18.29 ± 1.29 cm/s, 6.70 ± 0.91 cm/s, and 0.70 ± 0.02, respectively. Our findings were slightly divergent, with 17.96 ± 1.07 cm/s, 6.57 ± 0.72 cm/s, and 0.64 ± 0.03. When we assessed the healthy group in both studies, a consistent trend of lower values was evident. Zelli et al. (2013) documented PSV at 22.71 ± 1.88 cm/s, EDV at 4.47 ± 0.47 cm/s, and RI at 0.81 ± 0.07, while our results were closely aligned at 22.29 ± 1.4 cm/s, 4.44 ± 0.33 cm/s, and 0.80 ± 0.02. 

When comparing our findings to those of Zelli et al. (2013), various factors could account for the differences. The age, weight, and sample size of the dogs examined might affect the results. Moreover, our research incorporated a range of breeds, while Zelli et al. (2013) exclusively studied German Shepherds. Importantly, our study featured dogs with subclinical BPH, whereas the referenced study did not specify the clinical conditions of their subjects. The use of different ultrasound machines and software could also introduce discrepancies. However, in general, our findings consistently showed that dogs affected by subclinical BPH typically exhibit elevated vascular velocity parameters, particularly in the PSV, EDV, and RI measurements, when compared to healthy dogs. While most parameters presented statistical differences, the RI values in the subcapsular region were the exception (*p* > 0.05). Despite that, our observations also align with the studies published by Gunzel-Apel et al. in 2001 and Niżański et al. in 2020, where higher vascular velocities were documented in BPH-affected prostate glands in comparison to their healthy counterparts [37,38].

### 4.4. CPSE Values

In the current study, we discerned notable differences in serum CPSE levels between healthy dogs and those affected by subclinical BPH. The healthy group exhibited a mean CPSE serum level of 38.85 ± 14.55 ng/mL, spanning a range from 17.53 to 67.8 ng/mL. In stark contrast, dogs with BPH showed a considerably elevated mean value of 203.3 ± 90.39 ng/mL, with levels ranging between 97.31 and 487.54 ng/mL (*p* < 0.001). Our findings align with prior research, such as that of Bell et al. (1995), which documented BPH dogs having appreciably higher CPSE concentrations (189.7 ng/mL) relative to their unaffected counterparts (41.8 ng/mL) [39]. Similar trends of elevated CPSE levels in BPH dogs have been substantiated in other studies as well [40,41,42]. Our results are in agreement with Alonge et al. (2018), who found that healthy dogs had average CPSE values of 38.9 ng/mL, while dogs with BPH had values averaging 184.9 ng/mL [22]. However, while the mentioned study proposed a clinical CPSE threshold of 52.3 ng/mL for BPH diagnosis via ROC analysis, our study, benefitting from a more robust sample size (n = 35 vs. n = 19), set forth a diagnostic threshold of 82.56 ng/mL for subclinical BPH using the same analytic method. The variations in these outcomes may be attributed to differences in CPSE analysis methods, sample sizes, and the inclusion of diverse canine male subjects in both studies. 

It is imperative to acknowledge that previous investigations have predominantly relied on Enzyme-Linked Immunosorbent Assay (ELISA) tests for the analysis of CPSE values [12,24,26]. In contrast, our study adopted a laser-induced fluorescence immunochromatographic analyzer, a method we deemed more reliable than ELISA. This assertion is supported by the findings of Navvabi et al. (2022), wherein the immunochromatographic test demonstrated high accuracy in diagnosing Hepatitis B surface antigen (HBsAg) in human blood, while the ELISA test exhibited acceptable sensitivity and specificity [43]. Consequently, we posit that our chosen method for CPSE analysis offers enhanced precision. However, further studies are needed to confirm or refute this statement, which constitutes the primary objective of our forthcoming study.

## 5. Conclusions

In conclusion, our study underscores the pivotal role of various diagnostic modalities, including rectal palpation, ultrasonography, and CPSE analysis, in the early detection of subclinical BPH in older intact dogs. By introducing novel thresholds for ultrasonographic features and CPSE values, our research enhances diagnostic accuracy, facilitating timely intervention. Furthermore, our innovative rectal palpation scoring system, in conjunction with CPSE and ultrasonography, significantly improves the precision of subclinical BPH identification. These diagnostic advancements advocate for the integration of comprehensive male screening programs into routine veterinary practice, enabling proactive identification and management of prostatic conditions. However, despite our results indicating that prostate gland volume is influenced by individual age and weight, it is crucial to acknowledge a limitation of our study: the absence of comprehensive lifetime breeding history for each participant. Nonetheless, early diagnosis and intervention remain essential for maintaining the overall health and reproductive well-being of intact dogs. Recognizing the significance of early detection underscores the need for alternative diagnostic modalities and supports the development of proactive and preventative treatment options. Our study strongly recommends the adoption of comprehensive prostate gland screening protocols, thereby safeguarding canine health and optimizing breeding status.

## Figures and Tables

**Figure 1 animals-14-01204-f001:**
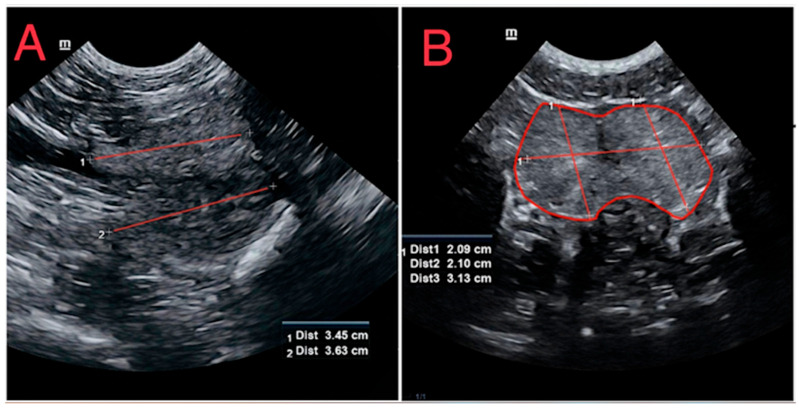
Ultrasonographic representation of measurements of the normal prostate gland in a 3-year-old Husky. (**A**): Longitudinal plane axis with red lines indicating measurements of prostatic length. (**B**): Transverse plane showcasing the prostate; the height and width are demarcated by red lines. The prostate gland is encircled in red, highlighting the region of interest and anatomical landmarks.

**Figure 2 animals-14-01204-f002:**
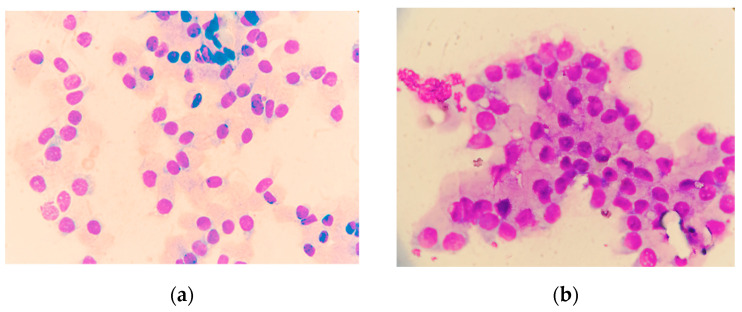
Cytological slides of prostate gland (magnification—100×). (**a**): normal prostate gland. Displayed are the acinar epithelial cells, organized in rows or small clusters. Note the distinct polarity of the epithelial cells, which are consistent in size. Their nuclei are round or slightly oval, featuring indistinct nucleoli and clear cytoplasm. (**b**): benign prostatic hyperplasia. High cellularity with epithelial cells forming large sheets. Cells may be columnar or polygonal, often lacking polarity. The nuclear/cytoplasm ratio remains similar to normal epithelial cells. Nuclei are predominantly consistent in size, slightly varying in morphology, and frequently exhibit a prominent nucleolus.

**Figure 3 animals-14-01204-f003:**
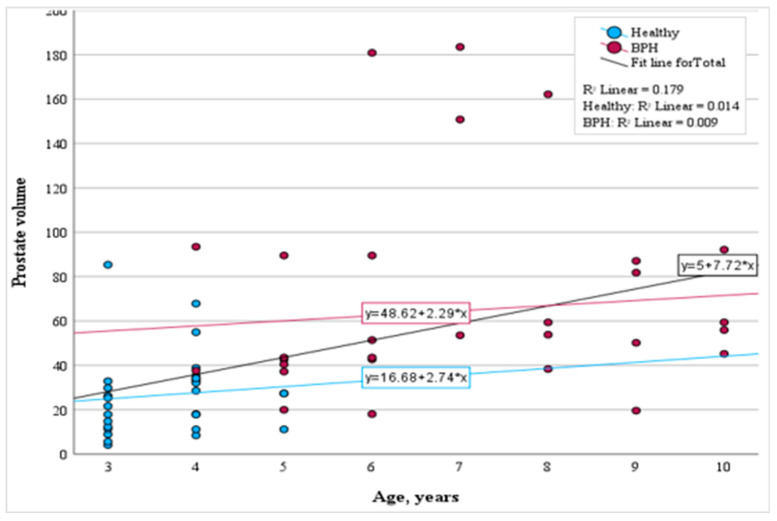
Scatter plot illustrating the correlation between age and prostate volume in distinct dog groups.

**Figure 4 animals-14-01204-f004:**
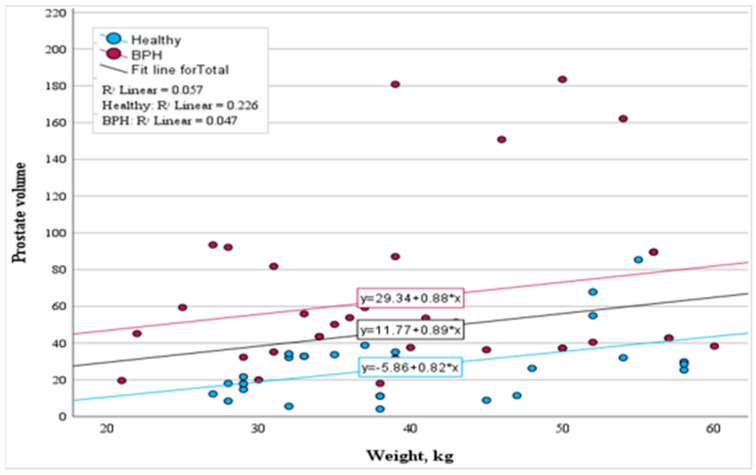
Scatter plot showcasing the correlation between dog weight and prostate volume.

**Figure 5 animals-14-01204-f005:**
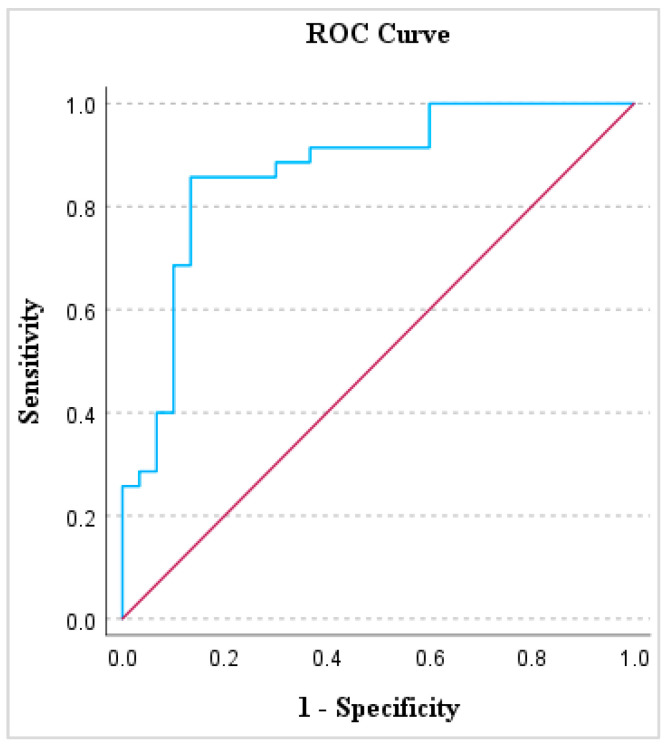
Receiver operating analysis (ROC) analysis of prostatic volume. The red line signifies the reference line, while the blue line depicts canine prostate-specific esterase (CPSE) values in ng/mL.

**Figure 6 animals-14-01204-f006:**
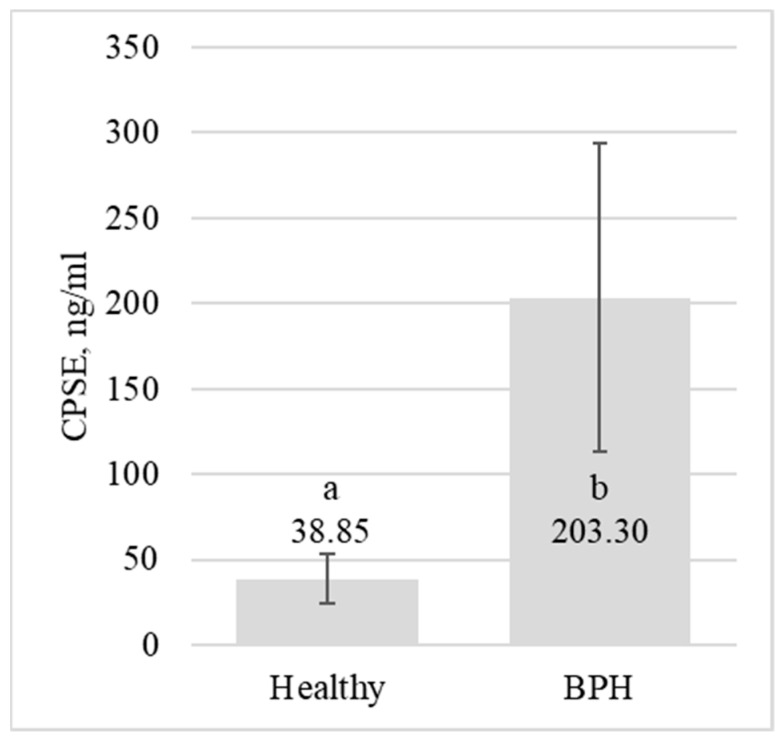
Comparison of mean CPSE values between the healthy and BPH-affected groups. The statistical significance, marked as ‘a’ and ‘b’, indicates a statistically significant difference between the study groups (*p* < 0.001).

**Figure 7 animals-14-01204-f007:**
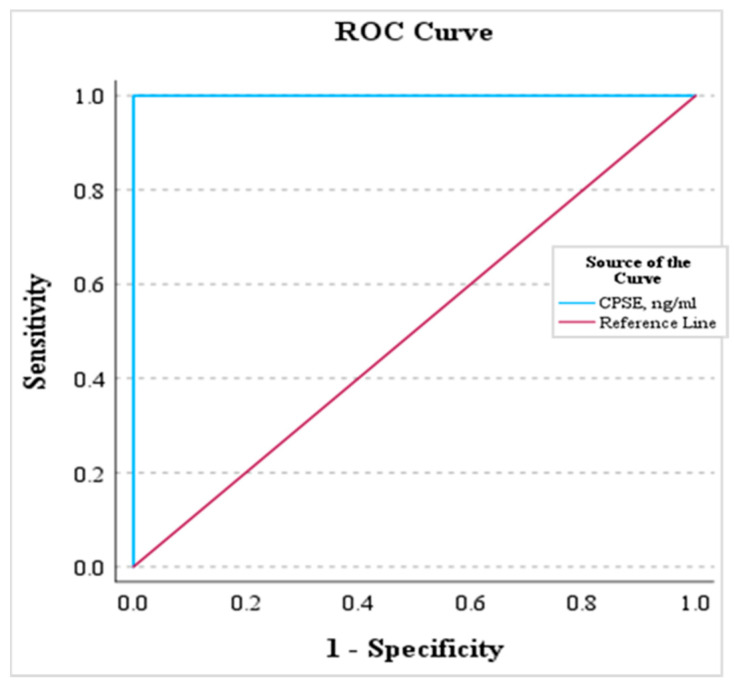
CPSE concentration explored through ROC analysis.

**Table 1 animals-14-01204-t001:** Results of the rectal palpation of the prostate gland in healthy and subclinical BPH-affected dogs.

Feature	Finding	Healthy (n = 30)	BPH (n = 35)	Statistical Significance
Shape	Symmetric	83.3% *	14.3% **	*p* < 0.001
	Asymmetric	16.7% *	85.7% **	
Consistency	Soft	83.3% *	31.4% **	*p* < 0.001
	Medium hard	16.7% *	54.3% **	
	Hard	0% *	14.3% **	
Urethral groove	Present	100% *	68.6% **	*p* < 0.001
	Absent	0% *	31.4% **	
Pain, score	0	63.3% *	0% **	*p* < 0.001
	1	26.7% *	31.4% *	
	2	10% *	54.3% **	
	3	0% *	14.3% **	
	4	0%	0%	
Size, score	0	30%	0%	*p* < 0.001
	1	60% *	34.3% **	
	2	10% *	65.7% **	
	3	0.0% *	0% **	
Position	Intra-pelvic	90.0% *	71.4% *	*p* = 0.115
	Extended caudally	10.0% *	20.6% *	
	Intra-abdominal	0% *	8.6% *	
Surface	Rough	86.7% *	45.7% **	*p* < 0.001
	Smooth	13.3%*	54.3% **	

*, **—proportions marked with different symbols in rows differed significantly (*p* < 0.05).

**Table 2 animals-14-01204-t002:** Means ± SD of prostatic artery Doppler parameters detected in marginal and subcapsular location in the normal group and subclinical BPH group.

Group	n	Marginal Location, PSV	Marginal Location, EDV	Marginal Location, RI	Subcapsular Location, PSV	Subcapsular Location, EDV	Subcapsular Location, RI
Healthy	30	22.29 ± 1.4 *	4.44 ± 0.33 *	0.80 ± 0.02 *****	15.36 ± 0.57 *	5.42 ± 0.55 *	0.65 ± 0.04 ***
BPH	35	34.1 ± 2.91 **	6.52 ± 0.86 **	0.81 ± 0.01 ******	17.96 ± 1.07 **	6.57 ± 0.72 **	0.64 ± 0.03 ****

PSV, peak systolic velocity; EDV, end diastolic velocity; RI, resistive index; symbols *; **—means marked with different letters in rows differed significantly (*p* < 0.001); ***; ****—means marked with different letters in rows differed insignificantly (*p* > 0.05); *****; ******—means marked with different letters in rows differed significantly (*p* < 0.01).

## Data Availability

All data presented in this study are available upon request from the corresponding author.

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
