# Peer review of "Comparative Evaluation of Diagnostic Methods for Subclinical Benign Prostatic Hyperplasia in Intact Breeding Male Dogs"

_animals, 2024, doi:10.3390/ani14081204_

Round 1

Reviewer 1 Report

Comments and Suggestions for Authors

The main question addressed by the research is the Detection and evaluation of subclinical prostatic hyperplasia with different non-invasive methods.

In my opinion, the results concerning canine prostatic-specific esterase are particularly interesting. They complement our knowledge about the levels of this enzyme in subclinical inflammation diseases of the prostate gland in dogs.

The authors introduce new thresholds for ultrasound features and CPSE values contribute to diagnostic precision.

I have no remarks regarding methodology. The study design and methodology are satisfactory.

In my opinion, the conclusions are consistent with the arguments presented in the paper. The results presented in this study can be easily implicated in the diagnosis of prostate gland which is essential for male dogs' veterinary practice. The methods presented in this study can simply help practitioners in the early detection of subclinical BPH in dogs.

The references are appropriate. Only 4 of 40 are published before 2000. Although we could suggest to Authors to make sure that they did not miss any papers published in last years in this area.

Tables and Figures are easy to understand and clearly illustrate the authors' workload. They present the results in a proper way.

This is interesting data according to canine andrology and satisfactory performed clinical paper, well prepared manuscript, easy to understand, with good study design and methodology. 

Due to the content of this study, evaluated paper is within the scope of the journal. In my opinion, this paper can be accepted for publication in Animals.

See the list of comments in the note for Authors. f.e.:

Line 49: well-being. In conclusion

Keywords: canine andrology, BPH, ultrasonography, rectal palpation, CPSE

Line 108: epithelium due

Line 136-138 – Please, add some notes that all dog patients had no testis tumors (diagnosed by ultrasound).

Line 185-186 Blood flow parameters  - uniform spelling in lower or capital letters

Author Response

Dear Reviewer,

Thank you very much for Your review of the present study article. The comments have been noted, and lines 49, 108, and 185-186 have been edited accordingly. Additionally, the keywords have been changed as per Your recommendation. A note in the Materials and Methods section regarding the absence of testicular tumors in the study dogs has been added, as You requested.

Once again, thank You very much.

Reviewer 2 Report

Comments and Suggestions for Authors

In the content of the article regarding rectal examination, information should be added that it is helpful to examine the prostate by palpating it from the abdominal cavity, before the pelvic symphysis, which allows it to be moved caudally and dorsally, and in some dogs, also assess the cranial and ventral part of the prostate   In this study, the authors proposed an innovative, although subjective, four-level scale method for assessing pain and prostate size and then performed a statistical analysis of these parameters. This analysis is presented only in table 1 without detailed commentary in the results and discussion sections   In line 108. no space or dot   The caption under Figure 3 is incorrect and states that it concerns Analysys CPSE concentration, and the content (line 286) shows that it concerns prostatic volume   In the work, the considerations concerned all medium and large dogs (included in one research group) without dividing them into smaller groups. This work may be an inspiration for future research in which the authors will analyze the parameters they study, taking into account age ranges, body weight and breed of dogs.   Due to the fact that benign prostatic hyperplasia often coexists with other prostate diseases (e.g. prostatitis), the work would gain in value if the authors in their research also took into account the results of blood tests enabling the diagnosis of possibly coexisting other prostate diseases. However, I believe that the adopted research model justifies presenting it in its current form with information on whether the aspect of coexistence, e.g. prostatitis, was taken into account. If so, such information should be included in the article

Author Response

Dear Reviewer,

Thank you very much for such a positive response regarding our study. Regarding your first note about palpating the prostate gland from the abdominal cavity in order to assess both sides, we will incorporate this into the text. Thank you for the suggestion.

Line 108 has been corrected, and the caption of Figure 3 has been amended as well.

We appreciate your attention to detail. Indeed, our future studies will focus on specific breeds, age and weight groups as well. This current study serves as an introduction to our forthcoming research.

Regarding the note about the absence or presence of prostatitis in the study dogs, we have duly noted your comment. Thank you for bringing it to our attention.

Overall, our study group would like to express our gratitude for your thorough review. All of your comments will be taken into consideration.

Reviewer 3 Report

Comments and Suggestions for Authors

This study describes the findings of 65 entire breeding dogs with BPH (n=35) and normal prostate (n=30) which were examined through rectal examination, ultrasonography and CPSE blood measurements.

The investigated BPH is a common and important prostate lesion, especially so for breeding dogs, and the performed analysis is informative for canine clinical practice as well as pathology.

In its current form, the study is however not yet suitable for publication. To improve the quality of the manuscript, the following comments need to be considered:

Major revisions

-        This study is about breeding dogs. Unfortunately, there is a complete lack of information about the actual breeding history of these dogs

-        Semen quality is essential for successful breeding and is directly influenced by the prostate functionality. There is however no information about the semen of the investigated dogs. Given the thoroughly performed prostate examination and measurement, investigation of the semen would have been important to include and would add value to the study in order to better understand the effect of BPH on breeding performance

-        The main aim as well as main conclusions are only vaguely defined.

-        Information about the examined dogs is poor. How were they selected? Were they examined exclusively for this study or did they present at the vet for other reasons (if yes, which?)? Which breeds were included?

-        Prostate dimension is one of the key parameters of this study. The size of the dogs obviously matters, however there is no mention of relative prostate size and about how much the type of dog (size, weight, breed, age) did influence the prostate dimension

-        It is not indicated if any of the examinations were performed in a blinded manner?

Minor revisions

-        Avoid ‘male’ throughout most of the manuscript as it is superfluent in a study about prostate

Simple summary

-        Line 25. Oversight is also common due to not routinely performing rectal examination

-        Line 27. CPSE does not indicate the actual method (presumably CPSE measuring in the blood)

-        Lines 27/28. ‘variations’ and ‘changes’ are too vague. What were they?

-        Line 40. For CPSE, indicate that it was measured in the blood

-        Line 40. CPSE defined as canine prostate specific esterase

-        Line 43. Distinct differences is too vague. Which were they?

-        Line 46. …various examinations, such as ultrasound, CPSE analysis, and…

-        Line 50. Delete ‘various’

Introduction

-        Line 55. … sex gland in the male reproductive tract of dogs.

-        Line 56/57. … and the overall health of canine patients.

-        Line 59. … this spontaneous lesion involves…

-        Line 67. … untreated, conditions such as prostatitis…

-        Line 67/68. … possibility of abscess formation can develop.

-        Line 70/71. … health of dogs with BPH, there can…

-        Line 74. … fact that BPH presents no clinical signs in many cases, …

-        Line 90. … abdominal organs, resulting in poor contrast images, affecting the visualization of the organ.

-        Line 93. …valuable method for examining…

-        Line 95. … challenges in practicality due to the necessity…

-        Line 99. … evaluation of the size, shape, structure and vascular…

-        Line 113. We aimed to define lesion-specific changes in the …

-        Line 114. … rectal palpation and to define a new scoring system.

Materials and Methods

-        Line 137. … examination of both testicles.

-        Line 143. … the prostate gland size, shape, consistency, surface, and position.

-        Rectal palpation scoring: the scoring systems are indicated to evaluate size and pain separately, however Score 3 and 4 includes both features.

-        Line 173. … highlight the borders…

Figure 1.

-        Line 192. … gland in a 3-years-old Husky.

Figure 2.

-        Line 208. … similar to normal epithelial cells.

-        Line 209. … consistent in size, slightly varying in …

Table 1.

-        The indicated features are rather subjective. If not defined more precisely, high inter observer variability is expected. Blinded examination would be very important to trust the findings as performed here.

-        The small letters ‘a’ and ‘b’ are confusing. Replace by commonly used symbols (e.g. *, **, ***)

-        Line 257. … displayed glandular asymmetry.

-        Line 260. … group demonstrated a tissue heterogeneity

-        Line 278. … prostate gland between both study groups…

Table 2.

-        As for Table 1, replace letters ‘a’, ‘b’ etc. by other defined symbols

Figure 4

-        Same as for Table 2

Comments on the Quality of English Language

Minor language editing is required.

Author Response

Dear reviewer,

Thank You very much for taking the time to review our manuscript. We have addressed the minor revisions as per Your suggestions. For detailed explanations of the major revisions, please refer to the attached file.

We kindly request Your favorable consideration of our study, particularly as it pertains to dogs with subclinical benign prostatic hyperplasia (BPH). We believe our research fills an important gap in the literature concerning this area, and we are eager to contribute to advancing knowledge in this field.

Once again, we appreciate Your time.

Round 2

Reviewer 3 Report

Comments and Suggestions for Authors

I would like to thank the authors for addressing most of the suggested points. The following revisions still need to be addressed:

-        Three parameters which are key when investigating prostate health and dimension as they have a direct effect on the gland’s structure and size, i.e. breed, age and breeding activity (e.g. how many years used as stud dog and how often mating) are either vaguely or not at all indicated for the studied dogs. If this information is not available, this needs to be clearly stated as a limitation of the study. If this information is available, the correlation of breed, age and sexual activity on the prostate size and structure would be important to include in this study

-        Add the reference PMID 28708525 when discussing the parameter age

-        Prostate dimension is one of the key parameters of this study. The size of the dogs obviously matters, however there is no mention of relative prostate size and about how much the type of dog (size, weight, breed, age) did influence the prostate dimension

Material and methods

-        Provide information about the studied dog breeds. The information (line 122) is too vague. I suggest including a table for this (could also be supplementary)

-        Provide information about the age of the studied dogs. The information (line 122) is too vague. I suggest including a table for this (could also be supplementary)

-        Dogs instead of ‘digs’ (line 147)

-        Information about the actual breeding history of the studied dogs is still missing. This is however an important aspect when investigating the prostate as it will influence the structure and health of the gland

-         

Author Response

Dear Reviewer,

Thank You for the second revision of the manuscript. The responses to all of Your revisions are in the attached file. We hope it meets Your expectations.
